# Remote monitoring of clubfoot treatment with digital photographs in low resource settings: Is it accurate?

Tracey Smythe[1]*, Marie-Caroline Nogaro[2], Laura J. Clifton[3], Debra Mudariki[4], Tim Theologis[2], Chris Lavy[5]

1 London School of Hygiene & Tropical Medicine, London, United Kingdom, 2 Paediatric Orthopaedic Surgery, Nuffield Orthopaedic Centre, Oxford University Hospitals NHS Trust, Oxford, United Kingdom, 3 South Tees Hospitals NHS Foundation Trust, James Cook University Hospital, Middlesbrough, United Kingdom, 4 Physiotherapy Department, Faculty of Health Sciences, University of Witwatersrand, Johannesburg, South Africa, 5 Nuffield Department of Orthopaedics Rheumatology and Musculoskeletal Science, University of Oxford, Oxford, United Kingdom

☯ These authors contributed equally to this work.
* tracey.smythe@lshtm.ac.uk

**Data Availability Statement:** All data files are available from the LSHTM database. The DOI for our datafiles titled 'ACT and photograph scores

## Abstract

### Background

Clinical examination and functional assessment are often the first steps to assess outcome of clubfoot treatment. Clinical photographs may be an adjunct used to assess treatment outcomes in lower resourced settings where physical review by a specialist is limited. We aimed to evaluate the diagnostic performance of photographic images of patients with clubfoot in assessing outcome following treatment.

### Methods

In this single-centre diagnostic accuracy study, we included all children with clubfoot from a cohort treated between 2011 and 2013, in 2017. Two physiotherapists trained in clubfoot management calculated the Assessing Clubfoot Treatment (ACT) score for each child to decide if treatment was successful or if further treatment was required. Photographic images were then taken of 79 feet. Two blinded orthopaedic surgeons assessed three sets of images of each foot (n = 237 in total) at two time points (two months apart). Treatment for each foot was rated as 'success', 'borderline' or 'failure'. Intra- and inter-observer variation for the photographic image was assessed. Sensitivity, specificity, positive and negative predictive values were calculated for the photographic image compared to the ACT score.

### Results

There was perfect correlation between clinical assessment and photographic evaluation of both raters at both time-points in 38 (48%) feet. The raters demonstrated acceptable reliability with re-scoring photographs (rater 1, k = 0.55; rater 2, k = 0.88). Thirty percent (n = 71) of photographs were assessed as poor quality image or sub-optimal patient position.

dataset' is https://doi.org/10.17037/DATA.
00001563

**Funding:** The authors received no specific funding
for this work

**Competing interests:** The authors declared that no
competing interests exist.

Sensitivity of outcome with photograph compared to ACT score was 83.3%–88.3% and
specificity ranged from 57.9%–73.3%.

## Conclusion

Digital photography may help to confirm, but not exclude, success of clubfoot treatment.
Future work to establish photographic parameters as an adjunct to assessing treatment out-
comes, and guidance on a standardised protocol for photographs, may be beneficial in the
follow up of children who have treated clubfoot in isolated communities or lower resourced
settings.

## Introduction

Clubfoot, or congenital talipes equinovarus (CTEV) is one of the most common musculoskele-
tal deformities seen at birth. The structural development of the bones and muscles of the foot
is affected, and the foot is fixed in a downward and inward position. Birth prevalence of club-
foot varies between 0.51 and 2.03/1000 live births in low and middle income countries (LMIC)
[1]. Approximately 80% of cases of clubfoot occur as an isolated birth defect and are termed
'idiopathic' as the cause is not known; the aetiology is multifactorial and it is likely that both
genetic and environmental factors are responsible [2]. If untreated the child develops a stiff
foot in a 'clubbed' shape, and has difficulty walking and participating in activities of daily liv-
ing, such as in school and play. However, more than 95% of cases are successfully treated with
the Ponseti method [3, 4]. This minimally invasive technique consists of two distinct phases,
the correction phase with manipulation, casting and often an Achilles tenotomy, and the main-
tenance phase with use of a foot abduction brace (FAB). In LMIC these cases are often man-
aged by clinical officers and physiotherapists [5–7] and input from orthopaedic surgeons may
be necessary to perform a percutaneous Achilles tenotomy for residual equinus deformity, or
if there is recurrence of the deformity following casting.

Elements of the deformity that recur are typically noted under clinical examination and
observation of function. The Assessing Clubfoot Treatment (ACT) score has recently been
shown to easily and reliably assess the results of CTEV deformity treated with the Ponseti
method in patients of walking age [8]. It was developed for clubfoot therapists to assess the
results of Ponseti treatment in children of walking age, in low resource settings, where access
to qualified therapists may be limited. The ACT score provides a simple to use indicator to
monitor the quality of clubfoot correction and may assist clinicians to determine their long-
term results, and identify which patients need onward referral for further intervention, such as
re-casting and/or surgery. The ACT score measures (i) passive range of dorsiflexion with knee
extended, (ii) whether the child can wear normal shoes (iii) and is pain free, and (iv) parent
satisfaction. One domain of CTEV assessment that is relevant alongside objective measures
and parent reported outcome measures, but has not yet been explored, is the use of digital pho-
tographs to visualise any residual deformity.

Access and use of digital photography, including in LMIC, is increasing. The use of smart-
phones has largely contributed to this, particularly in eye care [9–11]. In medicine, photo-
graphic data increasingly forms part of patients' medical records. Historically, medical
photography has been used to document rare or severe physical findings. Some surgical spe-
cialties have incorporated it into national and international guidelines, such as the British
Association of Plastic, Reconstructive and Aesthetic Surgeons guidelines in managing open

fractures, whereby absence of photographic documentation could be seen as adversely affecting a patient experience and outcome. For example, repeatedly taking down a dressing for multidisciplinary team assessment may result in unnecessary pain and risk of contamination [12]. In line with telemedicine, access to digital photography from a remote location without the need for in-person review allows for a wider access to expertise, which is especially useful in resource-limited settings. This is of particular interest in clubfoot treatment, where photographic evidence may enable repeated assessment and allow for multiple clinicians to comment on a foot deformity at a particular time point. In its simplest form, photographs of children's lower limbs and feet taken at various angles may be an adjunct used in assessing treatment outcomes in isolated communities or lower resourced settings where in-person physical review by a specialist is limited. For parents, appropriate and relevant photographs of their child's feet may provide a useful reminder of the deformity correction they should strive to maintain to prevent recurrence/relapse and could be used as marker of progression. Photographs may also serve as a way to motivate parents to continue with treatment as they can see the correction of the child's feet during the corrective phase.

We aimed to assess correlation between the ACT score and photographic assessment of a child's foot in determining treatment outcome (success or onwards specialist referral) for idiopathic CTEV treated with the Ponseti technique.

## Materials and methods

This study was conducted and reported according to established STARD (Standards for Reporting of Diagnostic Accuracy Studies) guidelines [13].

The participants were those included in a cohort study to develop the ACT score [8, 14]. They were children with a diagnosis of idiopathic CTEV corrected by the Ponseti method at Parirenyatwa Hospital, Harare, who attended follow-up review in January 2017, 3.5–5.0 years from initial casting.

### Data collection

Following informed parental consent, the ACT tool was administered. The question about the plantigrade position of the foot was answered first by independent physical examination of the child in the supine position by two physiotherapists, with the knee extended and through the measurement of passive range of dorsiflexion of the ankle joint. The remaining three questions of the ACT score were answered by the carers about the child's pain, ability to wear shoes and satisfaction (Table 1). We used the score to identify those patients who definitely need referral and further treatment (failure: score 8 or less) and those with a definite successful outcome (success: score 11 or more). Further discrimination is needed to decide how to manage patients with an ACT score of 9 or 10, and these scores were classed as 'borderline'.

**Table 1. ACT questions and score.**

| Score | 1.The foot is plantigrade | 2.Does your child complain of pain in their affected foot? | 3.Can your child wear shoes of your/their choice? | 4.How satisfied are you with your child's foot? |
|---|---|---|---|---|
| 0 | Does not reach plantigrade, with additional adduction, cavus or varus | Yes and it often limits their activity | Never | Very dissatisfied |
| 1 | Does not reach plantigrade, no additional deformity | Yes and it sometimes limits their activity | Sometimes | Somewhat dissatisfied |
| 2 | Plantigrade achieved | Yes but it does not limit their activity | Usually | Somewhat satisfied |
| 3 | More than plantigrade i.e. some dorsiflexion | No | Always | Very satisfied |

One physiotherapist then took digital photographs of the patient's lower limbs. Patients were bare-foot, and had their limbs exposed to above the level of the knee. Three sets of photographs were taken for each affected lower limb: i) standing view in the coronal plane from the back to assess the degree of residual hindfoot varus, ii) lateral view with the patient supine, knees extended and attempted passive dorsiflexion of the foot (by the second physiotherapist using either their hand or wooden plank) to assess for residual equinus deformity, and iii) lateral view with the patient standing with the knees flexed to assess for residual equinus deformity. Although we took three photographs, we were largely expecting results of images (ii) and (iii) to be similar given that isolated gastrocnemius contracture is not usually an issue with this pathology.

Two blinded orthopaedic surgeons viewed the digital photographs independently on a laptop computer and used goniometers to assess these angles on the digital photographs (S1 Fig). The residual cavus and midfoot adductor deformity that may be seen with a clubfoot deformity were not formally measured on these photographs. There were no exclusion criteria although a note was made if patient/foot position or quality of photograph was sub-optimal.

## Data management and analysis

All data were entered into a Microsoft Excel 2000 (Microsoft Inc., Redmond, Washington) software package. Data were analysed using Stata 14.1 (Stata-Corp 4905, Lakeway Drive College Station, Texas, 77845, USA.). Photographs were stored in a password protected drive.

The total ACT score was calculated within a range of 0 to 12 for each foot. The two blinded orthopaedic surgeons individually analysed and rated the series of photographs for each foot for 'success', 'borderline' and 'failure' on two separate occasions that were two months apart. 'Success' was defined as any combination of: valgus or neutral hindfoot alignment, plantigrade or dorsiflexion of the ankle joint with the knee extended, and plantigrade or dorsiflexion of the ankle joint with the knee flexed. 'Failure' was defined as any combination of: varus hindfoot alignment and plantarflexed ankle with the knee extended and not improved with the knee flexed. Any foot which did not clearly fit in these two categories was labelled as 'borderline'.

Descriptive statistics were used to characterise the study population. The degree of inter- and intra-rater correlation was assessed using kappa coefficient [15]. The treatment outcome score was calculated using the ACT tool (reference standard) and compared to the score based on photographic assessment of the residual foot deformity. We assessed the sensitivity and specificity of raters to detect either success or failure of clubfoot correction from digital photographs. The borderline score was combined with the success score to create a binary score (success or failure) in both the ACT and the photograph assessment. In combining these scores, we hypothesised that this would decrease false positive rate and therefore limit the number of unnecessary referrals, essential especially in resource poor settings.

## Ethical considerations

The Medical Research Council of Zimbabwe (MRCZ/B/789) and the London School of Hygiene & Tropical Medicine (LSHTM ref.: 11132) granted ethical approval. The caregiver provided informed written consent. Transport costs were reimbursed.

## Results

A total of 53 children (79 feet) from the initial cohort of 68 children had both an ACT score and digital photographs that were available for review. The first 15 patients of the ACT study did not have photographs taken of their feet as they were part of a pilot study. Table 2 summarises the patient characteristics.

**Table 2. Patient characteristics.**

| Characteristics | | N (%) |
|---|---|---|
| Gender | Male | 39 (74%) |
| | Female | 14 (26%) |
| Affected foot | Right | 39 (49%) |
| | Left | 40 (51%) |
| Average length of follow up | | 31.49 months (95%CI: 26.3–36.7 months) |

## Treatment outcome score using ACT score and digital photography

Forty-two (53%) CTEV feet had a successful outcome score following Ponseti treatment when assessed with the ACT score. About a quarter (n = 19, 24%) were assessed to have failed treatment and a similar number (n = 18, 23%) were assessed as 'borderline'. Although no patients were excluded from analysis, 21 (27%) (rater 1) and 29 (37%) (rater 2) feet were assessed as poor quality picture or sub-optimal patient position. When outcomes were assessed using digital photography, 42–56% of feet were graded as 'success', 24–30% as failure and 19–28% as borderline by both raters at two different time points (Table 3). There was perfect correlation between ACT score and raters 1 and 2 at both time-points in 38 feet (success n = 24 (57%), borderline n = 4 (22%), failure n = 10 (53%)).

Three quarters (n = 60, 76%) of feet assessed using the ACT score had a successful outcome when the borderline score was combined with the success score (Table 4). This proportion is closely mirrored by rater 1 at both time points, but rater 2 was shown to be slightly more conservative with 70% of feet being scored as successful based on the photograph alone.

## Rater agreement on outcomes from digital photograph

With regards the three scores of 'success', 'borderline' and 'failure', rater 2 demonstrated good reliability with rescoring (k = 0.88), and rater 1 was fair (k = 0.55). This was because rater 1 scored more feet as success in the second rating, with a decrease in borderline scores. There was good inter-observer reliability in the first assessment (k = 0.82), this decreased in the second (Table 5).

When the borderline score was combined with the success score to create a binary variable, the rater agreement improved (Table 6). The Intra- and inter-observer agreement was universally high ranging from 87.3% to 98.7%.

## Diagnostic accuracy

The comparison of digital photograph-based evaluation of clubfoot correction with the ACT score as reference is summarised in Table 7. The sensitivity was high for both raters and at

**Table 3. Clinical outcome according to ACT score and rater assessed digital photographs.**

| | ACT score n(%) | Photographic rating n(%) | | | |
|---|---|---|---|---|---|
| n = 79 feet (53 patients) | ACT score | Rater_1a* | Rater_2a* | Rater_1b** | Rater_2b** |
| Success | 42 (53.2) | 39 (49.4) | 35 (44.3) | 44 (55.7) | 33 (41.8) |
| Borderline | 18 (22.8) | 20 (25.3) | 21 (26.6) | 15 (19.0) | 22 (27.8) |
| Failure | 19 (24.0) | 20 (25.3) | 23 (29.1) | 20 (25.3) | 24 (30.4) |

* (a) first assessment in June

**(b) second assessment in August

**Table 4. Summary outcome of assessment for success and failure.**

| n = 79 feet (53 patients) | ACT score n(%) | Photographic rating n(%) | | | |
|---|---|---|---|---|---|
| | | Rater_1a* | Rater_2a* | Rater_1b** | Rater_2b** |
| Success | 60 (76.0) | 59 (74.7) | 56 (70.9) | 59 (74.7) | 55 (69.6) |
| Failure | 19 (24.0) | 20 (25.3) | 23 (29.1) | 20 (25.3) | 24 (30.4) |

(a) first assessment in June

(b) second assessment in August

both time points ranging from 83.3% to 88.3%. However, the ability of raters to correctly identify a foot that had failed treatment according to the ACT score and would benefit from onwards specialist referral was lower, in particular for rater 1 and specificity ranged from 57.9% to 73.7%. Similarly, the positive predictive value of photographs for successfully treated clubfeet is higher than the negative predictive value. A lower negative predictive value suggests that some of the patients with failure of clubfoot correction on image-based evaluation may have been referred unnecessarily.

## Discussion

There was perfect correlation between ACT score and both raters at both time-points in 38 (48%) feet between the three categories of 'success' (n = 24, 57%) 'borderline' (n = 4, 22%) and 'failure' (n = 10, 53%). This suggests that it is easier to recognise a clearly well-corrected foot and a foot with clear residual clubfoot deformity, as opposed to one which is partially improved (borderline) via digital photography. The raters demonstrated fair (rater 1, k = 0.55) to good (rater 2, k = 0.88) reliability with rescoring of photographs. Inter-observer reliability was high in assessing well-corrected clubfeet but lower when identifying sub-optimal correction. There was fair correlation between outcomes when evaluated by photograph and clinical assessment; when the two categories of 'success' and 'failure' were determined, sixty (76%) patients assessed using the ACT score had a successful outcome, and this was similar to evaluation of the photographs (n = 55–50, 70%–75%). Whilst all photographs were gradable, approximately 30% (n = 71/237) of the images were assessed as not ideal. The raters assessed 21 to 29 (27%–37%) photographs as poor quality picture or sub-optimal patient position.

Raters can identify patients that need to be referred from photographic assessment (sensitivity: 83.3%–88.3%). However, the ability of raters to correctly identify a foot that had failed treatment according to the ACT score was lower (specificity: ranged from 57.9% to 73.7%). Lower specificity means that patients are more likely to be referred unnecessarily. The positive predictive value of photographs for successfully treated clubfeet is higher than the negative predictive value, which means that a normal looking foot on photograph is unlikely to be assessed

**Table 5. Inter- and intra-observer reliability.**

| | Agreement % | Kappa (95%CI) |
|---|---|---|
| Intra-rater (1) | 72.15% | 0.55 (0.49–0.56) |
| Intra-rater (2) | 92.4% | 0.88 (0.79–0.90) |
| Inter-rater (a) | 88.61% | 0.82 (0.75–0.90) |
| Inter-rater (b) | 72.15% | 0.56 (0.47–0.68) |

(a) first assessment in June

(b) second assessment in August

**Table 6. Inter- and intra-observer reliability for success and failure.**

|  | Agreement % | Kappa (95%CI) |
|---|---|---|
| Intra-rater (1) | 87.3% | 66.5 (0.47–0.86) |
| Intra-rater (2) | 98.7% | 97.0 (0.91–1.00) |
| Inter-rater (a) | 93.7% | 84.1 (0.71–0.98) |
| Inter-rater (b) | 89.9% | 74.9 (0.59–0.91) |

(a) first assessment in June

(b) second assessment in August

to require onward referral for re-casting or surgery. Nevertheless, if the photograph suggests a foot with residual clubfoot deformity, this does not necessarily mean that onward referral was necessary when assessed with the ACT score. With the view of being resource efficient, this may limit the usefulness of the photographic evaluation by expert opinion remotely. However, patients with clubfoot may tolerate or adapt to residual deformity more readily when very young, but less so with age [16]. Early management in these cases may avoid worsening of residual or relapsed deformity that may become more refractory to correction in older children. The usefulness of photographic evaluation by a remote expert may prove particularly useful in these cases.

Ultimately, the aim of most foot and ankle interventions is to obtain a plantigrade, painless, "shoeable" foot [17]. Use of photographic assessment only to assess success of a treatment outcome may be simplistic, and the value of direct patient/parent feedback in terms of function and symptoms cannot be ignored. However, visual assessment of a residual deformity by a remote expert may provide some degree of correlation with the interpreted success of a treatment outcome. Remote expert opinion could be a useful screening adjunct in deciding which patients would benefit from onwards referral for a specialist opinion.

Our study has limitations. Determining the precise level of the ankle joint when measuring angles was occasionally confounded by poor lighting and a lack of contrast between the dark patient skin and dark background. In addition, due to suboptimal image quality or patient position, approximately a third of image-based evaluations were assessed by analysing only two rather than all three views, thus limiting the amount of data points available from which to draw conclusions. In addition, clubfoot deformity is a three-dimensional deformity, and our measurements only attempt to capture deformity in two planes. An additional limitation is the reduction of the definition of success or failure of a treatment to a combination of angular measurements, and more complex parameters may need to be considered when evaluating treatment outcome in clubfoot correction. With regard the cohort and calculation of the ACT score, children were not excluded if they had previously developed a relapse necessitating repeat cast treatment or surgery, and this may have influenced the clubfoot deformity and caregiver satisfaction.

Much work over the past two decades has focussed on developing clubfoot treatment programmes in LMIC, including education about the congenital deformity with a view to address

**Table 7. Diagnostic accuracy.**

| Rater Number (month) | Sensitivity (95%CI) | Specificity (95%CI) | Positive predictive value (95%CI) | Negative predictive value (95%CI) |
|---|---|---|---|---|
| 1 (June) | 88.3% (77.4–95.2) | 68.4% (43.4–87.4) | 89.8% (79.2–96.2) | 65.0% (40.8–84.6) |
| 2 (June) | 85% (73.4–92.9) | 73.7% (48.8–90.9) | 91.1% (80.4–97.0) | 60.9% (38.5–80.3) |
| 1 (Aug) | 85.0% (73.4–92.9) | 57.9% (33.5–79.7) | 86.4% (75.0–94.0) | 55.0% (31.5–76.9) |
| 2 (Aug) | 83.3% (71.5–91.7) | 73.7% (48.8–90.9) | 90.9% (80.0–97.0) | 58.3% (36.6–77.9) |

stigma that has long been associated with this condition. With increased community aware-ness, and children enrolled in such treatment programmes, it is imperative that these patients are followed-up appropriately with timely onward referral when necessary, and that patient drop-out or premature patient discharge is minimised. Based on our findings and study limita-tions, we cannot recommend using digital-based evaluation by remote specialist alone to determine the success or failure of Ponseti treatment. However, in combination with previ-ously documented scores and outcome measures (such as the ACT score), photographic review could be a useful adjunct in the evaluation. We recommend that if photographic images are to be used, particular attention is made to appropriate lighting, clear instructions for patient positioning (e.g. directly face on or sideways to the camera) and that an appropriate coloured background is used to facilitate visualisation of the ankle joint.

Ongoing research questions were framed by the gaps in evidence identified through this study. Future studies may seek to establish standardisation of all three photographs in the standing position to provide weighted views for assessment, which may assist raters in deter-mining the status of borderline scores. Studies that include photographs of the unaffected side may provide useful control information.

## Conclusion

Raters demonstrate acceptable reliability with rescoring photographs of children's feet that have been followed up after clubfoot treatment with the Ponseti method. There was fair corre-lation between outcomes when evaluated by photograph and clinical assessment, and this may be improved through the development of specific guidance to capture gradable photographs. Raters identify patients that need to be referred from photographic assessment with a high sen-sitivity, however the ability of raters to correctly identify a foot that had failed treatment according clinical assessment is lower. Further work is needed to determine whether other photographic parameters may be useful to remote experts to assist in determining outcome following Ponseti treatment. A larger number of raters and a standardised protocol in taking the photographs may be beneficial.

## Supporting information

**S1 Fig. Example of measuring clubfoot deformity from photographs.**
(DOCX)

## Author Contributions

**Conceptualization:** Tracey Smythe, Marie-Caroline Nogaro, Debra Mudariki, Chris Lavy.

**Data curation:** Tracey Smythe.

**Formal analysis:** Tracey Smythe, Marie-Caroline Nogaro, Laura J. Clifton, Debra Mudariki.

**Investigation:** Tracey Smythe, Debra Mudariki.

**Methodology:** Chris Lavy.

**Supervision:** Tim Theologis, Chris Lavy.

**Writing – original draft:** Tracey Smythe, Marie-Caroline Nogaro.

**Writing – review & editing:** Tracey Smythe, Marie-Caroline Nogaro, Laura J. Clifton, Debra Mudariki, Tim Theologis, Chris Lavy.

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
