## [Decision Letter · Decision Letter 0]

23 Mar 2020

PONE-D-20-03165

Remote monitoring of clubfoot treatment with digital photographs in low resource settings: is it accurate?

PLOS ONE

Dear Dr Smythe,

Thank you for submitting your manuscript to PLOS ONE. After careful consideration, we feel that it has merit but does not fully meet PLOS ONE’s publication criteria as it currently stands. Therefore, we invite you to submit a revised version of the manuscript that addresses the points raised during the review process.

We would appreciate receiving your revised manuscript by May 07 2020 11:59PM. To enhance the reproducibility of your results, we recommend that if applicable you deposit your laboratory protocols in protocols.io, where a protocol can be assigned its own identifier (DOI) such that it can be cited independently in the future. For instructions see: http://journals.plos.org/plosone/s/submission-guidelines#loc-laboratory-protocols

We look forward to receiving your revised manuscript.

Kind regards,

James G. Wright

Academic Editor

PLOS ONE

Journal Requirements:

3. We note you have included a table to which you do not refer in the text of your manuscript. Please ensure that you refer to Table 3 in your text; if accepted, production will need this reference to link the reader to the Table.

Reviewers' comments:

Reviewer's Responses to Questions

**Comments to the Author**

1. Is the manuscript technically sound, and do the data support the conclusions?

Reviewer #1: Yes

Reviewer #2: Yes

2. Has the statistical analysis been performed appropriately and rigorously? 

Reviewer #1: I Don't Know

Reviewer #2: Yes

3. Have the authors made all data underlying the findings in their manuscript fully available?

Reviewer #1: Yes

Reviewer #2: Yes

4. Is the manuscript presented in an intelligible fashion and written in standard English?

Reviewer #1: Yes

Reviewer #2: Yes

5. Review Comments to the Author

Reviewer #1: The authors sought to assess correlation between the Assessing Clubfoot Treatment (ACT) score and photographic assessment of Ponseti-treated clubfeet. ACT scores were obtained for each child. They were classified as success or failure (borderline scores were combined with success). Photos were taken of 79 feet and these were reviewed by 2 blinded orthopaedic surgeons who rated the feet as success if they saw valgus or neutral alignment of the hindfoot and plantigrade or dorsiflexion of the ankle joint. All others were deemed as failure. These ratings were then compared to the ACT scores. Comparing photograph-based evaluation with the ACT scores, the authors found the sensitivity was between 83.8% and 88.3%, while the specificity was between 57.9% and 73.3%. They noted that the ability of raters to correctly identify a foot that had failed treatment according to the ACT score was low. The authors concluded that photos may help confirm, but not exclude, success of clubfoot treatment; further work was needed to standardize the how the photos were taken and establish photographic parameters to more accurately assess treatment outcomes from the photos.

Thank you for allowing me to review this excellent manuscript. I have the following questions for the authors:

1. Introduction section, line 85: How was the ACT score validated?

2. Methods section: Am I correct to assume none of the patients in the cohort developed a relapse necessitating repeat cast treatment or surgery as this may influence patient symptoms and caregiver satisfaction?

3. Methods section, line 137: It might he helpful to the reader to include an example of the 3 good sets of photographs. Is there evidence in the literature to suggest differences in tightness of the soleus and gastrocnemius muscles In Ponseti-treated clubfeet? I have not found this; perhaps you have.

4. Results section, line 179: Were the ACT scores and photographic outcomes calculated for each ‘child’ or ‘foot’? Gray et al. (Clin Orthop Relat Res (2014) 472:3517–3522) found that patients with bilateral clubfeet have a high correlation between right and left feet for baseline severity and response to intervention. Accordingly, they argued that results in 2 limbs of the same patient do not represent independent observations. Could their observation have impacted your study?

5. Results section, line 184: Should this read Table 3?

6. Discussion section, paragraph beginning on line 235: You seem to infer a patient with a residual or relapsed clubfoot deformity—but who has no symptoms or shoe wear problems, and high caregiver satisfaction—would not benefit from repeat cast treatment or surgery to prevent the later onset of symptoms? Is there evidence to support this viewpoint? Children may tolerate or adapt to residual deformity more readily when very young, but less so with age. One may argue that it would be better to manage a residual or relapsed deformity early because it is likely to worsen and may become more refractory to correction in older children. If the latter argument is correct, then the usefulness of photographic evaluation by a remote expert may prove even more useful than you have concluded.

Reviewer #2: The research topic is timely and the study is worthy of publication. In general, the methodology is sound and the conclusions appropriately derive from the research.

Under question #3 above, a url needs to be appended to the LSHTM database reference in accordance wit the stated requirement

I submit the following comments in the interests of helping the authors improve the quality of the publication.

There needs to be some minor revisions as follows:

In the abstract the authors state that this is a prospective, single center cohort study. I question that it is prospective. The title of the research indicates that it is a diagnostic accuracy study. It includes two study periods of the same images to evaluate intra-observer validity, but this does not make it prospective. I note that it is "piggybacked" on a research study by the same group published in the Journal of Foot and Ankle Research, reference #8 in the reference section. The patient cohort in this study is drawn from the cohort in the previous study. That publication is a level II diagnostic study. I suggest this study is the same and the word "prospective" needs to be deleted from the abstract.

In the methodology the authors used three digital photographs of the patient's lower limbs, two of which are in standing position and one supine with passive dorsiflexion. These are demonstrated in the appendix. It is stated that the passive dorsiflexion with knee in extension and the weighted dorsiflexion with knee flexed are to simulate the Silverskjold test for contracture of the gastrocnemius muscle. To be comparative these two tests should be done in similar manner, either supine unweighted, or weighted. Since the crux of the research relies on measurements of the submitted photographs, the authors need to clarify in the text why they chose their technique. Of relevance is the known discrepancy between measured dorsiflexion passively and weighted, with weighted dorsiflexion giving increased angles. (See Baggett & Young, J Am Podiatr Med assoc. 1993 May;83(5):251-4).

It would have been more clear to standardize all three photographs in the standing position. I wonder if some of the difficulties in establishing the "borderline" cluster of patients came from discrepancy between these two images (see lines 160/161). The authors are encouraged to go back over their data and see if discrepancy between the weighted and unweighted dorsiflexion views were problematic in evaluating the classification. They are encouraged to discuss the choice of position and possible impact on the final result in the text.

Line 143 states "the lateral views were repeated for the other leg if there was bilateral foot involvement". Was the standing review in the coronal plane from the back to assess the degree of residual hind foot varus not done?

Also, in unilateral cases, were photographs taken of the non-involved side? Data gleaned from the normative side would have been useful as a control particularly since actual goniometric measurements were performed and not just a visual evaluation of plantigrade or not.

I found the results section difficult to read and had to read it several times to make sense of it. I am sure other readers will have the same difficulty. This section should be rewritten for clarity purposes.

• There should be better segregation of the ACT score and photographic scores for clarity. This should also apply to the tables, where more clear distinction between the act score: and photographic columns could be made.

• Line 184 incorrectly refers to table 2 for the referenced data. I am presuming the reference is for table 3.

• The digital photography data presented in line 183-185 is not reflected in any of the tables despite the erroneous reference to table 2. There must be alignment in the text and the submitted tables.

• Line 181-183 elaborates on those patients who had poor quality pictures or suboptimal patient position. How is this data reflected in the submitted tables? Were these patients excluded, or put into the borderline category? Evaluation of this group should be clearly identified in the text and the tables. The limitation section, line 255, indicates that fully a third of images were assessed with only two of the three views. How were the evaluations made in these circumstances?

• In the methodology section, lines 46 to 48, the methodology for obtaining the photographs is discussed. It is not clear whether these images were taken by physiotherapists in a clinic situation or on outreach. At issue is how standardized the procedure was. There will obviously be differences in the standardized referral clinic environment as compared to outreach situations. Was the research done by clustering the subjects in a research clinic environment, or did it mirror the desired condition of performance in a non-specialized community scenario. A statement in this regard is relevant in order to understand why there were images of poor quality. This issue should also be elaborated in the discussion section since it impinges on the methodology of obtaining photographic images in the clinical scenario remote from a specialized center.

• Lines 192-196 elaborate on the inter-rater reliability. Comment is made that rater 2 was more conservative than rater 1 at 70%. The table shows 70.9 versus 74.7. At the second go round the percentages are 69.6 Versus 74.7 as expressed in table #4. The question is whether these differences are statistically significant? I recognize that the sample size is small, but it appears to me that these are not significant differences.

One editing correction: Line 88 utilized “DF” without prior clarification of “dorsiflexion”.

The submitted paper does contribute useful information to the literature and I encourage the authors to evaluate the above recommendations before final submission.

6. PLOS authors have the option to publish the peer review history of their article (what does this mean?). If published, this will include your full peer review and any attached files.

Reviewer #1: Yes: Lewis E. Zionts

Reviewer #2: No

---

## [Author Response · Author response to Decision Letter 0]

2 Apr 2020

We have included a letter of response in our attachments, with formatting to assist the reader. 

Below is our unformatted response:

PONE-D-20-03165

Remote monitoring of clubfoot treatment with digital photographs in low resource settings: is it accurate?

PLOS ONE

We thank the reviewers and the Academic Editor for their comments and suggestions and for assisting in the improvement of our manuscript. Our responses are bolded below, with changes to the manuscript underlined. The line numbers refer to the lines on the track changes version. 

Reviewer #1: The authors sought to assess correlation between the Assessing Clubfoot Treatment (ACT) score and photographic assessment of Ponseti-treated clubfeet. ACT scores were obtained for each child. They were classified as success or failure (borderline scores were combined with success). Photos were taken of 79 feet and these were reviewed by 2 blinded orthopaedic surgeons who rated the feet as success if they saw valgus or neutral alignment of the hindfoot and plantigrade or dorsiflexion of the ankle joint. All others were deemed as failure. These ratings were then compared to the ACT scores. Comparing photograph-based evaluation with the ACT scores, the authors found the sensitivity was between 83.8% and 88.3%, while the specificity was between 57.9% and 73.3%. They noted that the ability of raters to correctly identify a foot that had failed treatment according to the ACT score was low. The authors concluded that photos may help confirm, but not exclude, success of clubfoot treatment; further work was needed to standardize the how the photos were taken and establish photographic parameters to more accurately assess treatment outcomes from the photos.

Thank you for allowing me to review this excellent manuscript. I have the following questions for the authors:

1. Introduction section, line 85: How was the ACT score validated?

Thank you for your review of our manuscript. The ACT score has been shown to be reproducible and reliable, providing the user with consistent results. The diagnostic accuracy test for the ACT score demonstrates that it is measuring what the user is expecting it to measure (ie whether further intervention is required or not) and therefore validity. However, additional studies are required to provide this external validation and these are ongoing. We have therefore included the following information about the ACT score:

Line 76-78 The Assessing Clubfoot Treatment (ACT) score has recently been shown to easily and reliably assess the results of CTEV deformity treated with the Ponseti method in patients of walking age [8].

2. Methods section: Am I correct to assume none of the patients in the cohort developed a relapse necessitating repeat cast treatment or surgery as this may influence patient symptoms and caregiver satisfaction?

The patients in the cohort were reviewed 3.5 to 5 years after initial casting and no patients were excluded if they had experienced a relapse that required either repeat cast treatment of surgery. As our assessment was based on the current position of the foot, we have included this information in the limitations section of the discussion. 

Line 280 – 283 With regard the cohort and calculation of the ACT score, children were not excluded if they had previously developed a relapse necessitating repeat cast treatment or surgery and this may have influenced the clubfoot deformity and caregiver satisfaction.

3. Methods section, line 137: It might be helpful to the reader to include an example of the 3 good sets of photographs. Is there evidence in the literature to suggest differences in tightness of the soleus and gastrocnemius muscles In Ponseti-treated clubfeet? I have not found this; perhaps you have.

We provide an example of the sets of photographs and measurements in S1 Appendix and have re-titled this in line with PLOS ONE requirements, and to assist the reader.

Line 155 – 157 Two blinded orthopaedic surgeons viewed the digital photographs independently on a laptop computer and used goniometers to assess these angles on the digital photographs (S1 Fig).

Following your query, our understanding of your question is that it relates to our sentence:

Photographs ii) and iii) were used together as a surrogate measure of Silfverskiöld test [15].

We have removed this sentence as we were largely expecting results of images (ii) and (iii) to be similar given that isolated gastrocnemius contracture is not usually an issue with this pathology and we have included this in line 151 - 154:

Line 151 – 154 Although we took three photographs, we were largely expecting results of images (ii) and (iii) to be similar given that isolated gastrocnemius contracture is not usually an issue with this pathology.

4. Results section, line 179: Were the ACT scores and photographic outcomes calculated for each ‘child’ or ‘foot’? Gray et al. (Clin Orthop Relat Res (2014) 472:3517–3522) found that patients with bilateral clubfeet have a high correlation between right and left feet for baseline severity and response to intervention. Accordingly, they argued that results in 2 limbs of the same patient do not represent independent observations. Could their observation have impacted your study?

Thank you for your request for clarification. We calculated the ACT scores and photographic outcomes for each foot, and we have amended the sentence on line 145:

Line 145: Three sets of photographs were taken for each affected lower limb: i) standing views in the coronal plane from the back to assess the degree of residual hindfoot varus, ii) lateral view with the patient supine, knees extended and attempted passive dorsiflexion of the foot (by the second physiotherapist using either their hand or wooden plank) to assess for residual equinus deformity, and iii) lateral view with the patient standing with the knees flexed to assess for residual equinus deformity.

We agree with Gray et al. (2014) that pooling clinical results of patients who present with bilateral clubfeet is statistically inappropriate when assessing response to intervention. The authors recommend using a random effects model to assess the association between potential predictors and the outcome, and this random effects model means that the right and left feet on the same child were not treated as independent. However, we believe that as our focusses on diagnostic accuracy between two outcomes, that this observation would have minimal impact on our study. 

5. Results section, line 184: Should this read Table 3?

Thank you. We have replaced the wording (now as line 196) to read Table 3. 

6. Discussion section, paragraph beginning on line 235: You seem to infer a patient with a residual or relapsed clubfoot deformity—but who has no symptoms or shoe wear problems, and high caregiver satisfaction—would not benefit from repeat cast treatment or surgery to prevent the later onset of symptoms? Is there evidence to support this viewpoint? Children may tolerate or adapt to residual deformity more readily when very young, but less so with age. One may argue that it would be better to manage a residual or relapsed deformity early because it is likely to worsen and may become more refractory to correction in older children. If the latter argument is correct, then the usefulness of photographic evaluation by a remote expert may prove even more useful than you have concluded.

Thank you for your suggestion. We have included this additional information from line 263 – 270, and have also included a recent reference to support younger children with clubfoot tolerating and adapting to residual deformities more readily than older children.

Line 263 – 270 With the view of being resource efficient, this may limit the usefulness of the photographic evaluation by expert opinion remotely. However, patients with clubfoot may tolerate or adapt to residual deformity more readily when very young, but less so with age [17]. Early management in these cases may avoid worsening of residual or relapsed deformity that may become more refractory to correction in older children. The usefulness of photographic evaluation by a remote expert may prove particularly useful in these cases.

Reviewer #2: The research topic is timely and the study is worthy of publication. In general, the methodology is sound and the conclusions appropriately derive from the research.

We thank you for your review and suggestions.

Under question #3 above, a url needs to be appended to the LSHTM database reference in accordance wit the stated requirement

Thank you. We agree to provide the doi for the LSHTM database reference should our manuscript be accepted for publication. 

I submit the following comments in the interests of helping the authors improve the quality of the publication.

There needs to be some minor revisions as follows:

In the abstract the authors state that this is a prospective, single center cohort study. I question that it is prospective. The title of the research indicates that it is a diagnostic accuracy study. It includes two study periods of the same images to evaluate intra-observer validity, but this does not make it prospective. I note that it is "piggybacked" on a research study by the same group published in the Journal of Foot and Ankle Research, reference #8 in the reference section. The patient cohort in this study is drawn from the cohort in the previous study. That publication is a level II diagnostic study. I suggest this study is the same and the word "prospective" needs to be deleted from the abstract.

Thank you for your suggestion. We have deleted the word ‘prospective’ from the abstract, and include ‘diagnostic accuracy’ to assist the reader.

Line 36 – 37 In this single-centre diagnostic accuracy study, we included all children with clubfoot from a cohort treated between 2011 and 2013, in 2017.

In the methodology the authors used three digital photographs of the patient's lower limbs, two of which are in standing position and one supine with passive dorsiflexion. These are demonstrated in the appendix. It is stated that the passive dorsiflexion with knee in extension and the weighted dorsiflexion with knee flexed are to simulate the Silverskjold test for contracture of the gastrocnemius muscle. To be comparative these two tests should be done in similar manner, either supine unweighted, or weighted. Since the crux of the research relies on measurements of the submitted photographs, the authors need to clarify in the text why they chose their technique. Of relevance is the known discrepancy between measured dorsiflexion passively and weighted, with weighted dorsiflexion giving increased angles. (See Baggett & Young, J Am Podiatr Med assoc. 1993 May;83(5):251-4).

Following your query, our understanding of your question is that it relates to our sentence:

Photographs ii) and iii) were used together as a surrogate measure of Silfverskiöld test [15].

We have removed this reference as highlighted by reviewer 1 and have replaced it with the following information:

Line 151 – 154 Although we took three photographs, we were largely expecting results of images (ii) and (iii) to be similar given that isolated gastrocnemius contracture is not usually an issue with this pathology.

It would have been more clear to standardize all three photographs in the standing position. I wonder if some of the difficulties in establishing the "borderline" cluster of patients came from discrepancy between these two images (see lines 160/161). The authors are encouraged to go back over their data and see if discrepancy between the weighted and unweighted dorsiflexion views were problematic in evaluating the classification. They are encouraged to discuss the choice of position and possible impact on the final result in the text.

Thank you for your recommendation. We agree that potentially standardising photographs to a weighted/standing position may assist with establishing the scores of clubfeet that were determined as ‘borderline’ and we have included a final paragraph on future study recommendations and provide this additional in lines 308-311.

Line 308-311 Ongoing research questions were framed by the gaps in evidence identified through this study. Future studies may seek to establish standardisation of all three photographs in the standing position to provide weighted views for assessment, which may assist raters in determining the status of borderline scores.

Line 143 states "the lateral views were repeated for the other leg if there was bilateral foot involvement". Was the standing review in the coronal plane from the back to assess the degree of residual hind foot varus not done?

Thank you for your request for clarification. We have removed this sentence, and changed the wording at the beginning of the paragraph to explain that three views were taken for each limb that was affected: 

145-146 Three sets of photographs were taken for each affected lower limb

Also, in unilateral cases, were photographs taken of the non-involved side? Data gleaned from the normative side would have been useful as a control particularly since actual goniometric measurements were performed and not just a visual evaluation of plantigrade or not.

In unilateral cases photographs were only taken of the involved side. We agree that data from the normative side would have been useful as a control and have included this in our recommendations section. 

Line 311 – 312 Studies that include photographs of the unaffected side may provide useful as a control.

I found the results section difficult to read and had to read it several times to make sense of it. I am sure other readers will have the same difficulty. This section should be rewritten for clarity purposes.

• There should be better segregation of the ACT score and photographic scores for clarity. This should also apply to the tables, where more clear distinction between the act score: and photographic columns could be made.

Thank you for your suggestion. We have included footnotes for ‘a’ and ‘b’, which relate to the timeframe (were a was assessment in June and b was assessment in August) and we believe will assist the reader. We also provide thicker borders to delineate between the ACT score and photographic scores, and additional headings of ‘ACT’ score and ‘photographic rating’ in Tables 3 and 4 (Line 205 and Line 218). 

• Line 184 incorrectly refers to table 2 for the referenced data. I am presuming the reference is for table 3.

We have corrected the reference to read ‘Table 3’

• The digital photography data presented in line 183-185 is not reflected in any of the tables despite the erroneous reference to table 2. There must be alignment in the text and the submitted tables.

Lines 183-185 relate to a subset of information in Table 3. We have therefore moved this information to assist the reader in understanding that the data are provided within Table 3. 

Line 197 – 201 When outcomes were assessed using digital photography, 42 - 56% of feet were graded as ‘success’, 24 – 30% as failure and 19 – 28% as borderline (Table 2) by both raters at two different time points (Table 3). There was perfect correlation between ACT score and raters 1 and 2 at both time-points in 38 feet (success n=24 (57%), borderline n= 4 (22%), failure n=10 (53%)).

• Line 181-183 elaborates on those patients who had poor quality pictures or suboptimal patient position. How is this data reflected in the submitted tables? Were these patients excluded, or put into the borderline category? Evaluation of this group should be clearly identified in the text and the tables. The limitation section, line 255, indicates that fully a third of images were assessed with only two of the three views. How were the evaluations made in these circumstances?

One of our aims for this paper was to evaluate if assessment of photographs was possible, and to what extent. Although no patients were excluded from analysis, 21-29 feet (depending on the observer) were assessed without the data from photographic view (iii) due to poor quality picture or sub-optimal patient position. Both surgeons still graded these feet but using 2 rather than all 3 pictures. In general, the surgeons found it easier to calculate the amount of dorsiflexion on picture (ii). They suggest that this may be due to the fact that the leg and foot was held by a physiotherapist and they weren’t relying on the child to follow instructions so the positioning was more reliable. We therefore suggest recommendations for use:

Line 134 – 137 We recommend that if photographic images are to be used, particular attention is made to appropriate lighting, clear instructions for patient positioning (e.g. directly face on or sideways to the camera) and that an appropriate coloured background is used to facilitate visualisation of the ankle joint.

• In the methodology section, lines 46 to 48, the methodology for obtaining the photographs is discussed. It is not clear whether these images were taken by physiotherapists in a clinic situation or on outreach. At issue is how standardized the procedure was. There will obviously be differences in the standardized referral clinic environment as compared to outreach situations. Was the research done by clustering the subjects in a research clinic environment, or did it mirror the desired condition of performance in a non-specialized community scenario. A statement in this regard is relevant in order to understand why there were images of poor quality. This issue should also be elaborated in the discussion section since it impinges on the methodology of obtaining photographic images in the clinical scenario remote from a specialized center.

Thank you for your suggestion. The images were taken by physiotherapists in a clinic situation, and this particular tertiary referral hospital had variable lighting and regular electricity cuts throughout the day. It therefore mirrors the context in non-specialised community scenarios. We encourage clinicians to attend to these challenges, particularly those of lighting, in lines 304 - 307

Line 304 – 307 We recommend that if photographic images are to be used, particular attention is made to appropriate lighting, clear instructions for patient positioning (e.g. directly face on or sideways to the camera) and that an appropriate coloured background is used to facilitate visualisation of the ankle joint.

• Lines 192-196 elaborate on the inter-rater reliability. Comment is made that rater 2 was more conservative than rater 1 at 70%. The table shows 70.9 versus 74.7. At the second go round the percentages are 69.6 Versus 74.7 as expressed in table #4. The question is whether these differences are statistically significant? I recognize that the sample size is small, but it appears to me that these are not significant differences.

We agree that the sample size is small and therefore it would not be appropriate to undertake tests of statistical significance. We have therefore changed the sentence to read that rater 2 was slightly more conservative:

Line 210 – 213 This proportion is closely mirrored by rater 1 at both timepoints, but rater 2 was shown to be slightly more conservative with 70% of feet being scored as successful based on the photograph alone.

One editing correction: Line 88 utilized “DF” without prior clarification of “dorsiflexion”.

Thank you. We have corrected the sentence to read ‘dorsiflexion’

Line 86 – 87 re-casting and/or surgery. The ACT score measures (i) passive range of dorsiflexion with knee extended

The submitted paper does contribute useful information to the literature and I encourage the authors to evaluate the above recommendations before final submission.

We thank you for your review and for assisting with the improvement of our manuscript.

---

## [Decision Letter · Decision Letter 1]

24 Apr 2020

Remote monitoring of clubfoot treatment with digital photographs in low resource settings: is it accurate?

PONE-D-20-03165R1

Dear Dr. Smythe,

We are pleased to inform you that your manuscript has been judged scientifically suitable for publication and will be formally accepted for publication once it complies with all outstanding technical requirements.

With kind regards,

James G. Wright

Academic Editor

PLOS ONE

Additional Editor Comments (optional):

Reviewers' comments:

Reviewer's Responses to Questions

**Comments to the Author**

1. If the authors have adequately addressed your comments raised in a previous round of review and you feel that this manuscript is now acceptable for publication, you may indicate that here to bypass the “Comments to the Author” section, enter your conflict of interest statement in the “Confidential to Editor” section, and submit your "Accept" recommendation.

Reviewer #1: All comments have been addressed

Reviewer #2: All comments have been addressed

2. Is the manuscript technically sound, and do the data support the conclusions?

Reviewer #1: (No Response)

Reviewer #2: Yes

3. Has the statistical analysis been performed appropriately and rigorously? 

Reviewer #1: (No Response)

Reviewer #2: Yes

4. Have the authors made all data underlying the findings in their manuscript fully available?

Reviewer #1: (No Response)

Reviewer #2: Yes

5. Is the manuscript presented in an intelligible fashion and written in standard English?

Reviewer #1: (No Response)

Reviewer #2: Yes

6. Review Comments to the Author

Reviewer #1: (No Response)

Reviewer #2: Revisions have addressed questions posed in the initial review. i am happy that the paper can be submitted.

7. PLOS authors have the option to publish the peer review history of their article (what does this mean?). If published, this will include your full peer review and any attached files.

Reviewer #1: No

Reviewer #2: No

---

## [Editor Report · Acceptance letter]

4 May 2020

PONE-D-20-03165R1 

Remote monitoring of clubfoot treatment with digital photographs in low resource settings: is it accurate? 

Dear Dr. Smythe:

I am pleased to inform you that your manuscript has been deemed suitable for publication in PLOS ONE. Congratulations! Your manuscript is now with our production department. 

With kind regards,

on behalf of

Professor James G. Wright 

Academic Editor

PLOS ONE